# A Food Frequency Questionnaire for Hemodialysis Patients in Bangladesh (BDHD-FFQ): Development and Validation

**DOI:** 10.3390/nu13124521

**Published:** 2021-12-17

**Authors:** Shakil Ahmed, Tanjina Rahman, Md Sajjadul Haque Ripon, Harun-Ur Rashid, Tasnuva Kashem, Mohammad Syafiq Md Ali, Ban-Hock Khor, Pramod Khosla, Tilakavati Karupaiah, Zulfitri Azuan Mat Daud

**Affiliations:** 1Department of Food Technology and Nutrition Science, Noakhali Science and Technology University, Sonapur 3814, Bangladesh; shakil.ftns@gmail.com (S.A.); riponftns@gmail.com (M.S.H.R.); 2Institute of Nutrition and Food Science, University of Dhaka, Dhaka 1000, Bangladesh; tanjina.mili@gmail.com; 3Kidney Foundation Hospital and Research Institute, Dhaka 1216, Bangladesh; rashid@bol-online.com (H.-U.R.); tasnuva.kashem@gmail.com (T.K.); 4Department of Nutrition Sciences, Kulliyyah of Allied Health Sciences, International Islamic University Malaysia, Kuantan 25200, Pahang, Malaysia; syafiqali@iium.edu.my; 5Faculty of Food Science and Nutrition, Universiti Malaysia Sabah, Kota Kinabalu 88400, Sabah, Malaysia; khorbanhock@gmail.com; 6Department of Nutrition and Food Science, Wayne State University, Detroit, MI 48202, USA; aa0987@wayne.edu; 7School of Biosciences, Taylors’ University, Subang Jaya 47500, Selangor, Malaysia; tilly_karu@yahoo.co.uk; 8Department of Dietetics, Faculty of Medicine and Health Sciences, Universiti Putra Malaysia, Serdang 43400, Selangor, Malaysia; 9Research Center of Excellence (RCoE) Nutrition and Non-Communicable Diseases, Faculty of Medicine and Health Sciences, Universiti Putra Malaysia, Serdang 43400, Selangor, Malaysia

**Keywords:** food frequency questionnaire (FFQ), hemodialysis, dietary assessment, chronic kidney disease (CKD), Bangladeshi food intake

## Abstract

Diet is a recognized risk factor and cornerstone for chronic kidney disease (CKD) management; however, a tool to assess dietary intake among Bangladeshi dialysis patients is scarce. This study aims to validate a prototype Bangladeshi Hemodialysis Food Frequency Questionnaire (BDHD-FFQ) against 3-day dietary recall (3DDR) and corresponding serum biomarkers. Nutrients of interest were energy, macronutrients, potassium, phosphate, iron, sodium and calcium. The BDHD-FFQ, comprising 132 food items, was developed from 606 24-h recalls and had undergone face and content validation. Comprehensive facets of relative validity were ascertained using six statistical tests (correlation coefficient, percent difference, paired *t*-test, cross-quartiles classification, weighted kappa, and Bland-Altman analysis). Overall, the BDHD-FFQ showed acceptable to good correlations (*p* < 0.05) with 3DDR for the concerned nutrients in unadjusted and energy-adjusted models, but this correlation was diminished when adjusted for other covariates (age, gender, and BMI). Phosphate and potassium intake, estimated by the BDHD-FFQ, also correlated well with the corresponding serum biomarkers (*p* < 0.01) when compared to 3DDR (*p* > 0.05). Cross-quartile classification indicated that <10% of patients were incorrectly classified. Weighted kappa statistics showed agreement with all but iron. Bland-Altman analysis showed positive mean differences were observed for all nutrients when compared to 3DDR, whilst energy, carbohydrates, fat, iron, sodium, and potassium had percentage data points within the limit of agreement (mean ± 1.96 SD), above 95%. In summary, the BDHD-FFQ demonstrated an acceptable relative validity for most of the nutrients as four out of the six statistical tests fulfilled the cut-off standard in assessing dietary intake of CKD patients in Bangladesh.

## 1. Introduction

Chronic kidney disease (CKD) is a major public health problem worldwide, in which it is estimated that 5–10 million people die annually due to kidney diseases [1]. Globally, about 8–16% of people are living with CKD [2,3], and the overall prevalence of CKD in Bangladesh alone is 22.48% [4], which is higher than the global rate [5]. Malnutrition occurs commonly in the CKD population [6], and a limited study has indicated that poor nutritional status is evident in HD patients in Bangladesh [7]. Although healthcare prioritizes medical treatment for CKD patients in Bangladesh [8], medical nutrition therapy is an area that is largely neglected.

Nutritional assessment is an integral component in the nutrition care process necessary for medical nutrition therapy. However, there is a lack of dietitians with specialized training to perform nutritional assessments and management of renal patients, particularly in low- to middle-income countries such as Bangladesh [9,10,11]. In this context, the diet for CKD patients must be individualized by considering the nutritional needs at various stages of CKD, treatment modalities, physical and psychological conditions as well as comorbidities experienced by these patients [12,13,14]. Periodic dietary assessment of CKD patients to improve their diet-related clinical outcomes is also necessary [15].

Several common methods for assessing dietary intake in epidemiological study settings include 24-h dietary recalls, weighed food records (WFRs), food frequency questionnaires (FFQs), and dietary history, although each presents with several limitations in assessing habitual dietary intake due to random and systematic errors in measurement [16]. While the WFR is often regarded as the reference method in assessing usual nutrient intake, this method is time-consuming and requires strict adherence to methodological requirements, which often is challenging to CKD patients, causing underreporting [16,17]. Relevant to the CKD population, the KDOQI Clinical Practice Guidelines recommend the use of the 3-day food record as a means to assess dietary intake [18]. However, its use imposes a disproportionately high burden on respondents (requiring literacy and high motivation, multiple days to assess usual intake, and induced changes to diet in repeated measures) and also requires trained nutrition professionals [19]. In contrast, the FFQ appears to be more practical to apply in the CKD/dialysis population [20], particularly in low-resource countries [21,22].

A food frequency questionnaire (FFQ) is a specific list of foods and beverages with a frequency response section to indicate how often each food item is consumed within a certain period of time [23]. There are three types of FFQs: qualitative, semi-quantitative, and quantitative. Qualitative FFQs have no additional information about portion size, while semi-quantitative FFQs collect information about portion size and quantitative FFQs collect information about typical portion size by using realistic food models or by providing pictures of various portion sizes [16]. As the food items are culturally specific, FFQs should always be developed and validated in the target population [23]. FFQ development involves a specific approach in choosing foods, developing background questions, and designing the frequency response section [16,24].

Three validation studies have been reported for the non-CKD population in Bangladesh—a 42-item dish-based semi-quantitative FFQ for rural areas [25], a 9-item semi-quantitative FFQ for the Health Effects of Arsenic Longitudinal Study (HEALS) [26], and a FFQ adapted from HEALS for cardiovascular risk assessment, with added food items [27]. The foods listed in these FFQs may not reflect the habitual intake of dialysis patients as this population are subjected to kidney-specific dietary restrictions and also have specific dietary needs [20]. There is no published FFQ specifically designed for advanced CKD patients undergoing hemodialysis (HD) treatment in Bangladesh (to the best of our knowledge). Therefore, the current study’s goal is to (a) develop a dialysis-specific FFQ for HD patients in Bangladesh (abbreviated as BDHD-FFQ) based on dietary data collated over a one-year period and (b) determine its relative validity with 3-day 24-h recalls and corresponding nutritional biomarkers.

## 2. Materials and Methods

### 2.1. Study Design

This study was structured in three phases [22]. In Phase I, we focused on developing the BDHD-FFQ. In Phase II, the BDHD-FFQ prototype was progressed to face and content validation, and in Phase III, the relative validation of the newly established BDHD-FFQ was carried out. The study flow is represented in Figure 1. The ethical boards of the Kidney Foundation Hospital and Research Institute, Bangladesh (KFHRIB) and Wayne State University, USA (IRB #123314M1F), approved the study protocol.

### 2.2. Phase I: Development of BDHD-FFQ

The construction of the BDHD-FFQ at the first stage of this study included dietary data (3-days of 24-h dietary recall (3DDR)—one weekend day, a dialysis day, and a non-dialysis day) [28] collected from maintenance hemodialysis (MHD) patients who participated in a clinical trial, namely, Palm Tocotrienols in Chronic Hemodialysis (PATCH) (https://clinicaltrials.gov/ct2/show/NCT02358967, accessed on 16 August 2021) over a period of one year. The dietary data was then utilized for the development of an initial dietary databank. The concept for the BDHD-FFQ was adapted from the HD-FFQ that was developed for Malaysian HD patients [22].

#### 2.2.1. Patients Recruitment

Patients (HD) of Bangladeshi ethnicity/origin, aged at least 18 years old, non-transplant, dialyzed for at least 3 months, and fit for assessment (no physical or mental disability) were included in the PATCH study. Only patients who gave their consent for participation were recruited. Dietary data (24-h dietary recalls) were obtained from a cohort of 102 HD patients at baseline, followed by 69, 56, and 48 patients at the subsequent 3-monthly encounter. The number of patients was reduced during the 3-monthly encounter due to dropouts (sudden death, transfer to another dialysis center, kidney transplantation, and reluctance to continue participation in the study).

#### 2.2.2. Dietary Data Analysis

Overall, a total of 621 24-h recalls were obtained from the PATCH cohort between September 2018 until September 2019, whilst only 606 recalls were used to establish the database and develop a food list in this study (15 recalls were excluded due to incomplete records). Five steps were followed to identify the foods that would be included in the BDHD-FFQ, as previously described [29]:(a)All the food items obtained from 24-h dietary recalls (606 recalls) were listed in Microsoft Office Excel 2016.(b)A total of 4517 food items were listed, which were further classified into 16 food groups (Cereals and Products; Cooked Rice; Fish, Shellfish and Products; Poultry Meat and Products; Vegetables; Pulses, Legumes, and Products; Milk and Dairy Products; Bakery Products and Sweets; Fruits; Snacks and Finger Foods; Traditional Pitha; Fast Food Chain; Soup; Noodles and Pasta; Beverages; and Miscellaneous).(c)Energy and nutrients relevant to the CKD population (i.e., protein, fat, carbohydrate, sodium, calcium, iron, potassium, and phosphate) were identified for all food items based on the existing Bangladesh Food Composition Database [30](d)Some food items were merged according to their similarity and the amount of nutrients per serving. This process leads to only 203 food items from a total of 4517 items.(e)The most representative food items (i.e., contributing up to 90% of dietary energy and nutrients of interest in the HD population’s food intake) were selected, and the residual 10 percent of the foodstuff were excluded [29].

The method provided us with a relative percentage of the impact of individual food items in the diet to the nutrient of interest, which is given by the formula:(1)$$\mathrm{Relative}\;\mathrm{contribution}\;\mathrm{of}\;\mathrm{the}\;\mathrm{item}\;\left(\%\right)=\frac{total\;nutrient\;of\;the\;item,\;N}{total\;nutrient\;of\;all\;the\;food}\times 100$$(where *N* represents the food item).

Finally, a total of 132 food items were short-listed within the predetermined 16 food groups.

#### 2.2.3. Development of a Nutrient Composition Database

Energy and nutrition information for single food items were mainly obtained from the Food Composition Table (FCT), representing the Bangladesh nutrients database [30]. The nutrient information data from other databases [31,32,33,34] were utilized if any food substance was not available in the FCT. If nutrient values for specific food items did not exist in the FCT, we acquired weighed recipes, and with the aid of a group of local nutritionists, we constructed a standard recipe to complete the nutrient information using Nutritionist Pro™ software (Axxya Systems LLC, Stafford, TX, USA). As the nutrient values reported in the FCTs were based on uncooked food, a verified yield factor for cooked food items was included to convert the cooked weight into the uncooked weight for the item. Yield parameters were either collected from the FCT of Bangladesh or computed by weighing the meals before and after cooking using a conventional and average recipe. In the case of commercial food items, nutritional information labeling was obtained when such information was unavailable in the food composition database (e.g., cake, biscuits). Similar food items were chosen as an alternative and incorporated into analyses for foods that had no nutrient record or where the recipe could not be established or the nutritional information was not labeled.

In addition to creating a food nutrient database, information pertaining to frequency of consumption and size of serving was built into the FFQ. Food consumption frequency was differentiated into four levels of intake (daily, weekly, monthly, and rarely), as previously described [22]. Food portion sizes were obtained from participants using standard household measures. The preliminary version of the BDHD-FFQ prototype comprised of 132 food items categorized into 16 food groups (Appendix A).

### 2.3. Phase II: Face and Content Validation

Validation for the face and content of the BDHD-FFQ was carried out with 12 HD patients (amateurs) and 12 “specialists”, encompassing nutritionists (*n* = 6) and nephrologists (*n* = 6), as previously described [35]. The amateurs enrolled were aged >18 years old, literate, and were non-healthcare persons. Content validation was conducted through face-to-face and online methods. For the face-to-face method, an expert panel meeting was organized, and then the researcher facilitated the content validation process. For the online method, an accessible online content validation form was sent to the specialists, with clear guidelines to assist the content validation process. The complete fill-up time of the FFQ was noted, and participants were requested to provide their feedback using a five-point Likert scale (1 = very poor, 2 = poor, 3 = fair, 4 = good, and 5 = very good) on standard aspects, as previously described [22]. The given scores were used to calculate the value for the Scale–Content Validity Index (S-CVI) [36]. Open comments from the specialists and amateurs were listed and documented. The snapshot of finalized BDHD-FFQ is available as a Appendix A.

### 2.4. Phase III: Relative Validity of the BDHD-FFQ

#### 2.4.1. Patients’ Recruitment

A cross-sectional study was carried out in four HD units, including government hospitals and private dialysis units from four districts, namely Dhaka, Chittagong, Mymensingh, and Comilla. A total of 116 patients were recruited by convenience sampling between February to April 2021. The inclusion criteria were: patients over the age of 18 who have been receiving regular HD treatment twice/thrice weekly for at least 3 months, as well as written consent to participate in the study. Patients who were unwilling to follow the study protocol, had compromised mobility, were hospitalized at the time of data collection, or who were without available biochemical data at the time of the study were excluded. Patients involved in the previous two phases of the study were excluded for this phase. The participant’s information, including socio-demographic characteristics, biochemical data, anthropometric measurements, and dietary data, was collected through interviews and medical records and analyzed until July 2021.

#### 2.4.2. Serum Biomarkers

Patient biochemical profiles, which were conducted on a quarterly basis, were obtained from their respective medical records. Generally, midweek pre-dialysis blood samples were collected from patients after at least an 8 h overnight fast. Skilled dialysis nurses performed the blood collection procedures at the respective HD centers. These biomarkers were evaluated for use as an additional reference marker to test the validity of the BDHD-FFQ. The data collection was planned to coincide with routine biochemical analysis of the respective dialysis centers to maintain the interval between dietary and biochemical parameters within one week.

#### 2.4.3. Dietary Data Assessment

All participants were given a BDHD-FFQ. The BDHD-FFQ is an English-language questionnaire with a Bangla translation. The first section of the questionnaire gives patients detailed instructions on how to keep track of their food intake. The 3DDR method was used as a reference/standard method, as described for Phase 1 and elsewhere [37]. A total of 348 24-h recalls were collected from 116 patients recruited in Phase III. However, only data from 97 patients (291 24-h recalls) were used in the validation study. A total of 19 patients were excluded from the final analysis due to incomplete data (demographic/biochemical/BDHD-FFQ) (*n* = 16) and dietary data under-reporting (*n* = 3). For the BDHD-FFQ, participants were required to identify foods consumed, as per the foods listed for specific food groups, and record their usual food consumption frequency and serving size. During the assessment, trained interviewers used common household measures (plates, bowls, cups, and serving spoons) to facilitate participant recall of the quantity of food items eaten. The BDHD-FFQ was applied within a week after 3DDR collection. The 3DDR data comprise 24-h dietary recalls on a weekend and two weekdays (a dialysis day and a non-dialysis day). The time interval for the collection of 3DDR in Phase III of the study is within a week (1–3 days between recalls, depending on dialysis treatment schedule). The completed BDHD-FFQs were rechecked by a trained nutritionist for accuracy and completeness. Any incomplete information was additionally checked and verified with the patients by the interviewer.

#### 2.4.4. Analysis of Dietary Data from BDHD-FFQ and 3DDR

Nutritionist Pro™ software (Axxya Systems LLC, Stafford, TX, USA) was used to analyze the 3DDRs, which references the Food Composition Table (FCT) of Bangladesh [30,31] and also carries researcher-constructed food recipes, where the reported food items are not cited in any database, as described in Phase I of the study. The BDHD-FFQ data were presented in terms of food consumption frequency. The conversion factors for food consumption frequency were 1.00, 0.14, and 0.03 according to day, week, and month, respectively. The following formula was used to compute daily nutrient intake from the BDHD-FFQ [38].
(2)Daily nutrient intake=Conversion factor×number of intake×number of portion taken×weight of food per portion

Patient’s dietary intake data, as reported using the BDHD-FFQ, was entered into Excel 2016 software (Microsoft Office) to compute daily nutrient intake as per the formulated nutrient composition database developed in Phase 1 (Section 2.2.3).

#### 2.4.5. Misreporting

The Goldberg cut-off method was implemented to evaluate misreporting or implausible diet records based on patient’s physical activity level (PAL) of 1.4 (moderate to low physical activity level) and compared with the energy intake (EI) to basal metabolic rate (BMR) ratio, where the value of <0.8, 0.8–2.4 and >2.4 for the EI:BMR index was used for an indicator of under-reporting, acceptable reporting, and over-reporting, respectively [39]. The BMR of patients was determined using the Harris–Benedict equation [40].

### 2.5. Statistical Analyses

All statistical analyses were carried out using IBM SPSS^®^ (version 26.0). Missing values and outliers were screened out prior to the analysis. Continuous variables were presented as mean ± SD, and descriptive statistics for all the parameters were performed. Continuous variables normality was ascertained using the Kolmogorov–Smirnov test.

For BDHD-FFQ validation by amateurs and specialists in Phase II of the study, an independent sample *t*-test was carried out to determine the mean differences across the food groups. The Item-Content Validity Index (I-CVI) was carried out between expert members in scoring the food items [41]. Before calculating the CVI, the Likert score was applied as either 1 (score ≥ 4) or 0 (score < 4). The Scale-Content Validity Index (S-CVI) is the proportion of items on a scale that received a relevance score of ≥4 by all specialists [42]. The considerable score for *S-CVI* was ≥0.78 when >9 specialists were involved [43].

In Phase III of the study, the means (±SD) of the energy and nutrients assessed by the BDHD-FFQ and 3DDRs were calculated. The difference between the FFQs and 3DDRs was tested by a paired-samples *t*-test for the energy and nutrients of interest. Correlations were measured using an intraclass correlation coefficient for unadjusted data. The intraclass correlation coefficient test was performed to determine the strength of association between the BDHD-FFQ and 3DDRs by both energy-adjusted and age-, gender-, and BMI-adjusted data. For the validation of the dietary intake assessment method using correlation coefficients, the following criteria, as suggested by Lombard et al. [44,45], were used: good outcome (*r* ≥ 0.50), acceptable outcome (*r* = 0.20–0.49), and poor outcome (*r* < 0.20). Pearson’s correlation was computed between estimates of nutrient intake obtained from the BDHD-FFQ and 3DDRs with the corresponding serum biomarkers.

Cross-classification classifies participants’ nutrient intake into categories, for example, tertiles, quartiles, or quintiles, based on two dietary assessment methods [46]. Participants within the same category of classification of nutrients as per the two dietary assessment methods are classified as “correctly classified” and interpreted as a percentage value. The “grossly misclassified” term is used when the nutrients by two dietary assessment methods are in opposite categories, which is also interpreted as a percentage value, with the following criteria; good outcome is presumed when ≥50% of the participants are classified within the same category and ≤10% of participants are classified in differing categories [44]. This test reflects the agreement at the individual level, and it helps in the ranking of nutrient intake data relevant to research that focuses on diet–disease relationships [45].

The cross-quartile classification analysis method would clearly show the acceptance level in placing patients in the same or different tertile [45]. In the analysis, patients’ nutrient intake for each nutrient of interest is divided into four tertiles. By comparing the nutrient data between the BDHD-FFQ and 3DDRs, patients are classified into the same tertile, same and adjacent tertile (±1), or wrongly classified tertile.

The weighted kappa statistics method is used to assess the agreement (excluding chance) between the BDHD-FFQ and 3DDRs. Lombard et al. mentioned that the interpretation criteria for weighted kappas are good outcome (≥0.61), acceptable outcome (0.20–0.60), and poor outcome (<0.20) [44,45].

The Bland-Altman method was used to evaluate the agreement of the BDHD-FFQ with 3DDRs. It indicates the presence, direction, and extent of bias as well as the level of agreement between the two measures at the group level [45]. The limit of agreement (LoA) (95% confidence limits of the normal distribution) is computed as the mean difference ± 1.96 SD and represents overestimation and underestimation [47,48]. However, between the two dietary methods, it is expected that 95% differences are within the 95% LoA [49]. We used Pearson’s correlation coefficient to assess proportional bias between mean and mean difference for each selected nutrient, where the interpretation criteria are good outcome (*p*-value > 0.05) and poor outcome (*p*-value < 0.05) [44,50].

There is no “gold standard” statistical procedure for assessing the new FFQs [44,45]. A review of the statistical tests used to evaluate dietary assessment method validity showed that there is no “set” number of statistical tests required to define absolute validity [45]. Dietary assessment validation studies often used a combination of multiple statistical tests [45]. The interpretation outcomes were considered for both group and individual level validity [45]. The term “acceptable validity” dictates that more than half of the performed statistical tests fulfilled the provided cut-off standard for individual and group levels.

## 3. Results

### 3.1. Phase I: Development of the BDHD-FFQ

Socio-demographic and clinical data of the study cohort obtained for Phase I of the study is provided as a Appendix A.

### 3.2. Phase II: Face and Content Validation for the BDHD-FFQ

Out of a maximum score of five, the specialists and amateur groups provided overall mean rating scores of 4.38 and 4.41, respectively, for all the food groupings. The scorings from these validator groups were not significantly different (*p* = 0.56). For all 16 food groupings, the highest score was observed in the “Milk and dairy products” group for the specialists (mean score 4.52 ± 0.45) and the highest score was observed in the “Vegetables” and “Pulses, legumes and their products” from the amateurs, with mean scores of 4.55 ± 0.47 and 4.55 ± 0.40, respectively. The lowest score was seen in the “Traditional Pitha” group rated by the specialists (4.30 ± 0.32) and “Fast food chain” group rated by the amateurs (4.17 ± 0.49). The scores of both specialists and amateur groups were compared for each food grouping. There is no significant difference between the validating groups for each food group (all *p* > 0.05). The respective scores accomplished for each food group by the specialist and amateur groups are presented in Table 1.

An additional test of the Item-Content Validity Index (I-CVI) was performed between the specialist group’s members in scoring the food items. The score attained for the Scale Content Validity Index (S-CVI) was 0.9 (Appendix A), where a value >0.78 is considered an acceptable score, which shows decent agreement between the specialists in terms of the relevancy or applicability of the food items listed in the BDHD-FFQ.

The open comments from the validators are listed (Appendix A). The specialists and amateurs were asked about the aspects of the BDHD-FFQ that could be improved or the things that they liked about the newly developed tool. All their comments were compiled, and the BDHD-FFQ was improved accordingly before proceeding to validation with a HD population in Bangladesh.

### 3.3. Phase III: Relative Validity of the BDHD-FFQ

#### 3.3.1. Study Population Characteristics and Clinical Data

In total, 116 HD patients were enrolled for this phase. However, a total of 16 patients were excluded due to incomplete data and 3 subjects were identified as mis-reporters (under-reporters) and were excluded from the final analysis (Appendix A). A total of 97 eligible subjects, comprising 50.5% males, completed Phase III. The mean age of the subjects was 49.8 ± 12.3 years old, with 32.0% aged between 51 to 60 years old and an overall mean BMI of 23.2 ± 4.9 kg/m^2^; 38.1% of patients had completed secondary school education. Most patients (73.2%) were not working. The mean dialysis vintage was 43.6 ± 27.1 months. All socio-demographic parameters and clinical data of the Phase III participants are presented in Table 2.

#### 3.3.2. Validity Tests between BDHD-FFQ and 3DDR

a.Intraclass Correlations Coefficient of BDHD-FFQ with 3DDR

In our study, interclass correlation coefficients were used to measure the overall strength and direction of the correlation between the two different methods at the individual level. Overall unadjusted dietary data showed acceptable to good positive correlations (all *p* < 0.05) between the BDHD-FFQ and 3DDRs, as shown in Table 3. This is ranging from *r* = 0.30 for dietary iron (mg/day) to *r* = 0.66 for energy (kcal/day). When energy was adjusted, correlations of nutrients between the BDHD-FFQ and the 3DDRs ranged from *r* = 0.23 for dietary iron (mg/day) to *r* = 0.53 for dietary phosphate (mg/day). However, when adjusted for additional covariates such as age, gender, and body mass index (BMI), nutrients assessed with the BDHD-FFQ remained significantly correlated with 3DDR data, albeit with poor correlation values (*r* < 0.20).

b.Correlation coefficients between BDHD-FFQ and 3DDR with serum renal profiles

The correlations of the BDHD-FFQ and 3DDRs with serum biomarkers are provided in Table 4. The BDHD-FFQ showed a higher correlation with corresponding serum renal profiles of interest compared to 3DDRs. Energy intake assessed by the BDHD-FFQ bore a significant correlation (*p* < 0.01) with serum potassium (*r* = 0.47), serum phosphate (*r* = 0.43), serum creatinine (*r* = 0.56), and serum urea (*r* = 0.55), whilst no such correlations were observed with 3DDRs. Additionally, dietary carbohydrate and fat consumption evaluated by the BDHD-FFQ were significantly correlated for all serum biomarkers; however, these observations were not detected with 3DDR consumption data. Dietary protein and phosphate intakes assessed by the BDHD-FFQ were significantly correlated, including all serum profiles; however, these observations were not detected by 3DDR dietary data except with serum potassium.

c.Mean difference between nutrients calculated as per BDHD-FFQ and 3DDR

The paired *t*-test represents a group-level agreement between the two assessments (BDHD-FFQ vs. 3DDR), whilst the percentage mean difference between the two reflects group agreement (size and direction of error at the group level). The mean differences between dietary nutrients obtained using the BDHD-FFQ and 3DDRs are presented in Table 5. The percent of mean differences ranges from 11.0% for calcium to 33.9% for iron. All nutrients had acceptable percent mean differences (≤20%) except for sodium, iron, and phosphate. Paired-sample *t*-test analysis results for BDHD-FFQ and 3DDR data revealed that the intake comparison was statistically significant for all nutrients (*p* < 0.01), implying that there was no agreement at the group level, indicating poor validity.

d.Cross-classification and weighted kappa between nutrients derived by BDHD-FFQ and 3DDR

The quarterly categorization of nutrient distribution was used to assess the consistency of all nutrient consumption when cross-classifying individuals between BDHD-FFQs with 3DDRs (<10% of gross misclassification; see Table 6). For nutrients classified in the same tertile, iron is recorded as the lowest value (29.9%), while energy has the highest value (42.27%). The percentage of nutrients classified in adjacent tertiles (±1 tertile apart) ranged from 37.11% for fat to 47.42% for potassium. Regarding the nutrients that were grossly misclassified (defined as ±3 tertiles apart), the lowest was energy (2.06%) and the highest was iron (7.22%). The cross-classification test reflects the agreement at the individual level, while the opposite tertile (%) showed that all nutrients of interest had good validity at the individual level (≤10%).

The weighted kappa coefficient showing highest for energy (0.43, 95% CI: 0.30–0.55), and the lowest for iron (0.12, 95% CI: 0.02–0.27) (see Table 6). A result between 0 and 1 for the weighted kappa coefficient is commonly expected agreement (excluding chance) at the individual level. Negative numbers imply an agreement “worse” than can be predicted by chance alone, whereas values of zero or near to zero indicate “no more than pure chance” [42]. The weighted kappa showed acceptable validity agreement at the individual level for all nutrients with the exception of iron.

e.Bland-Altman analysis between BDHD-FFQ and 3DDR

Bland-Altman plots pictured the agreement of the BDHD-FFQ with 3DDRs (Figure 2). In view of the Bland-Altman index, a positive mean difference (mean bias) was apparent for all nutrients, and the percentage data points within the limit of agreement (mean ± 1.96 SD) are above 95% for energy, carbohydrates, fat, iron, sodium, and potassium, with the exception of protein, phosphate, and calcium (Table 7). Acceptable validity for the Bland-Altman method is that the 95% limit of agreement, including 95% of differences between the two dietary methods [50].

Pearson’s correlations between mean and mean difference were significant (*p* < 0.05) for carbohydrate, iron, and phosphate, indicating a skewed Bland−Altman plot, which denotes there is proportional bias. Meanwhile, no significant (*p* > 0.05) correlation was observed for energy, protein, fat, sodium, calcium, and potassium, indicating no proportionate bias was observed.

## 4. Discussion

The Bangladeshi hemodialysis food frequency questionnaire (BDHD-FFQ) was developed to assist healthcare practitioners in assessing food intake and identifying patients at risk of suboptimal nutrient intakes, requiring intervention. Given the prevailing issues in low-resource settings without clinical dietitians, it is critical to facilitate dietary intake assessments with a well-validated and appropriate FFQ. The 132-item BDHD-FFQ in our study, in general, was able to adequately estimate energy and nutrients of interest for this CKD population (i.e., carbohydrate, protein, fat, calcium, sodium, potassium, and phosphate) just like the 3DDRs, with the exception of iron. However, the BDHD-FFQ was easier to administer, without the need for specialized nutrition assessment skills.

The BDHD-FFQ provided categorization of specific food groups such as “fish and shell fish”, “meat and poultry”, and “vegetables and legumes” using local recipes such as “curry”, “shallow/deep fried”, “bhuna”, and “cooked with/without vegetables”, based on the composite-meal-based FFQ adopted in the Malaysian HD-FFQ [53]. Edible oils and seasonings specific to Bangladeshi food preparation included soybean/rice bran/mustard/sunflower/rapeseed oil or ghee, salt, turmeric, coriander, red chili, cumin powder, raw green chili, onion, ginger, and garlic or their pastes, and, often, a pinch of sugar, garam masala powder/paste, soya sauce, mustard seed, black pepper, raw tomato, or fresh coriander leaves. These ingredients contribute to significant phosphate, potassium, and sodium intakes and their inclusion in our recipe construction closely reflects the actual intake of a patient.

In terms of usability, approximately half an hour is needed to fill up this FFQ, which concurs with other useful FFQs, as reported [22,54,55]. The number of food items in the BDHD-FFQ falls within the range recommended for FFQ construction [24] and is acceptable to the experts. In terms of the formatting features in the FFQ design, both the healthcare experts and amateurs were in good agreement with the familiarity of food items, food portion size, relevance to dietary practice, and flow of questionnaire as well as the clarity of the questionnaire. Of note, the estimated S-CVI score was 0.91, indicating that the overall content validity of the newly established BDHD-FFQ was good [56]. The BDHD-FFQ was validated against the 3DDR, which is the standard recommended assessment approach [17,24], and each of these methods were validated against serum biomarkers in this study. We found there was a good correlation between methods of dietary assessment (unadjusted model) for energy and CHO and an acceptable correlation for protein, fat, sodium, calcium, iron, potassium, and phosphorus. In comparison, a similar range of correlation coefficients (*r*) was also observed, from 0.31 for iron to 0.67 for energy, in testing a semi-quantitative FFQ validated in both rural Bangladesh and a larger prospective study for rural and urban areas [26], as well as another FFQ applied to patients with cardiovascular disease [27]. In addition, all nutrients showed acceptable agreement after being adjusted for energy, with higher *r*-values found for phosphate (good agreement). Adjustment for covariates reduced the *r*-value for all nutrients to <0.20, albeit with a significant *p*-value, indicating the influence of these factors [16], in line with other studies [25]. However, it is important to note that the practice of covariate adjustment is not commonly reported in other studies [26,43,57,58].

In the HD population, malnutrition is highly prevalent and reported in many countries [59,60,61,62,63]. Dietary energy intake (DEI) and dietary protein intake (DPI) are the indices critical to reporting when checking the diets of patients for nutritional adequacy [64,65]. We found that the BDHD-FFQ correlated satisfactorily with the 3DDR with regards to energy (*r*) and was lower for protein (*r*) intakes. In addition, other micronutrients such as iron and calcium also indicated a correlation between the methods. However, the BDHD-FFQ did overestimate for energy and all nutrients at the group level, which is inherent to the use of FFQs [66]. Overestimation is attributed to patient fatigue, stress, monotonous eating habits, and misinterpretation of food portion sizes, which can reduce the accuracy of nutrient recall data [67,68,69].

The pathophysiology of micronutrients consumed in the diet is changed with kidney failure. Nutrients such as phosphorous and potassium are likely to be retained as the kidney fails to maintain a balance. The BDHD-FFQ demonstrated superior correlation compared to the 3DDR against the respective biomarkers as regards dietary phosphorous (*r* = 0.52 vs. *r* = 0.18) and potassium (*r* = 0.27 vs. *r* = 0.11) intakes. This finding is consistent with the Malaysian HD_FFQ study on which our FFQ is conceptualized [22]. In a population of children and adolescents, an FFQ application was shown to be effectively correlated to dietary intakes of vitamin C and calcium with the respective serum biomarkers [70]. Another perspective to consider regarding the use of serum biomarkers to validate the FFQ is the issue of systematic error [17,57]. Most dietary methods are associated with recall bias [17,58,71] as subjects rely on their memory to report food items as well as portion size “as best they can” [67,68,69]. Therefore, our intention in benchmarking both dietary methods against serum biomarkers serves the purpose of identifying which instrument is subject to systematic error. We found that the 3DDR could not correlate to the biomarkers of phosphorous and potassium, whereas this was not an issue with the BDHD-FFQ.

It is essential that the development of an FFQ should be synchronized with locally consumed food databases to be relevant to health management [72]. Furthermore, the food items in the database should reflect seasonal availability [23,25]. Seasonal fruits and vegetables (e.g., mango, guava, bhorta, aachar, sak, bhaji) contribute significant phosphate and potassium. These perspectives were considered in the development of the BDHD-FFQ as diet recalls are conducted 4 times over 12 months for the same patient, which allows the capturing of seasonal variations of habitual food consumption. Therefore, despite daily changes in micronutrient intake, the BDHD-FFQ effectively captured micronutrient intakes due to the extensive food list typically consumed by this population [17].

The ability of the BDHD-FFQ to replace 3DDRs for dietary assessment in a low-resource setting lacking renal dietitians, such as in Bangladesh, is very critical in ensuring patients are monitored to prevent malnutrition and hyper toxicities [4,7]. Cross-classification, as opposed to correlation coefficients, can provide considerably stronger and unbiased pictures of how well the FFQ performs [17]. We found that cross-quartile data agreement (<10% gross misclassification) was evident for all nutrients of interest in our evaluation of the BDHD-FFQ. This meant the ability of the FFQ and 3DDR methods to classify the same people into the same nutritional intake categories [73] was resonant with the BDHD-FFQ [25]. Meanwhile, in terms of weighted kappa statistics, acceptable agreement was observed for all concerned nutrients except iron (0.12) between the BDHD-FFQ and 3DDR approaches, which were almost identical to those reported in other studies, which took place in other countries, including Bangladesh [25,74,75,76,77]. However, Bland-Altman plots showed that within 95% LoA, percentage datapoints above 95% were observed for energy, carbohydrates, fat, iron, sodium, and potassium at the group level, which is expected [49]. It is important to iterate that a judgment of whether or not such limits are acceptable would depend on the clinical context [50,78]. The BDHD-FFQ showed overestimations of intake (positive mean bias) for energy and nutrients, which is similar to other studies [22,25,79,80]. Although the FFQ is known to overestimate energy and nutrients [67,81,82], lack of agreement is usually detected by a Bland-Altman plot [50].

Overall, based on the six statistical tests performed, three were tested to be valid at the individual level (ICC, cross-classification, and weighted kappa), while the remaining were vaid at the group level (paired *t*-test, percent mean difference, and Bland-Altman). At the individual level, all nutrients showed acceptable to good validity except for iron. Meanwhile, at the group level, paired *t*-tests were significant, indicating no agreement between the BDHD-FFQ and 3DDRs. As expected, Bland-Altman methods showed an overestimation for all nutrients, and above-95% differences (datapoints) were within the limit of agreement for energy, carbohydrates, fat, iron, sodium, and potassium. Sodium, iron, and phosphate had poor agreement, indicated by a >20% mean difference between the two methods. The different facets of validity achieved in this study is visualized in Appendix A.

This study had some limitations. We elected to compare the BDHD-FFQ with the 3DDR and acknowledge that both methods share recall bias with regard to patient memory. Ideally, the reference standard for validation should be weighed food records [83]. However, as patients on dialysis are constrained with dialysis procedures, weighed food records are impractical. A number of studies have also noted similar approaches to developing and validating their FFQs in this population [17,18,24,38]. An additional limitation is that we did not perform a reproducibility evaluation due to time constraints and logistic limitations arising from the Covid-19 pandemic. Ideally, a reproducibility study should be performed within a 1-week to 1-month interval [84], depending on the characteristics of the FFQ [85], by administering the same BDHD-FFQ to the same group of subjects. It is important to maintain a good reproducibility to ensure that the FFQ captures the true regular dietary intake rather than a random variation in response [84]. As reproducibility may dictate the reliability of the BDHD-FFQ, it is recommended that a future study should address this aspect prior to its utilization.

This study had some strengths. Firstly, the development of the BDHD-FFQ was the first initiative taken in Bangladesh to enable health professionals to perform the critical task of dietary assessment in the dialysis population. This will overcome the issues of the lack of trained dietitians in a low-resource setting as this tool can be easily utilized by non-nutrition professionals. Secondly, we derived a food listing for the BDHD-FFQ based on the 3DDRs collected four times over one year for the same subjects. This facilitated a wider scope of food item inclusions, such as seasonal vegetables and fruits. We used the Bangladeshi food composition tables [30,31] for analyzing the nutrient content of the food items [83], which ensured relevance to local food choices. An important feature of this BDHD-FFQ is the detailing of food preparation techniques, allowing it to better quantify specific nutrients associated with the cooking process. More importantly, the study validated the newly established BDHD-FFQ against 3DDRs and serum biomarkers. A published FFQ, in use in Bangladesh for the normal population, only validated methods of dietary assessment without any biomarkers [25,26]. Additionally, the BDHD-FFQ has been developed in two languages: English and Bangla, which minimizes the impact of language barriers and facilitates self-administration when used by both healthcare workers and patients.

Finally, the study had a comparatively high participation rate, has comprehensive data collected by trained personnel, and used a variety of all conceivable tools to estimate food intake amount and portion size.

## 5. Conclusions

To the best of our knowledge, this is the first validity study of an FFQ specific to Bangladeshi HD patients using serum biomarkers and dietary assessments. This BDHD-FFQ demonstrated overall acceptable agreement for ranking individuals (when compared with 3DDR methods) to meet the recommended nutrition goals for HD patients. This newly developed BDHD-FFQ is a practical tool as (a) due to its convenient usage, as no specialized training is required for diet data collectors, (b) it can be self-administered for use in a large population, (c) it allows for a large temporary catchment (for months) and, therefore, is resistant to seasonal fluctuations, and (d) it provides a relatively good degree of validity in ranking items for each food item. We believe that this FFQ will help health care professionals in identifying patients with nutrient deficits and excess intake, allowing for individualized dietary intervention. However, we recommend that the BDHD-FFQ should be used with caution, considering the unavailability of reproducibility assessments and overestimation of nutrient intake, hence warranting a further evaluation to ascertain these findings.

## Figures and Tables

**Figure 1 nutrients-13-04521-f001:**
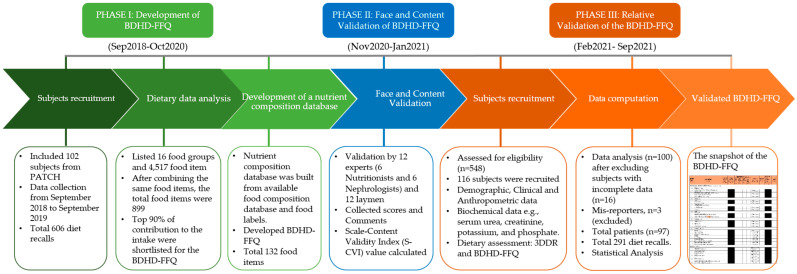
Development and validation process algorithm of the BDHD-FFQ.

**Figure 2 nutrients-13-04521-f002:**
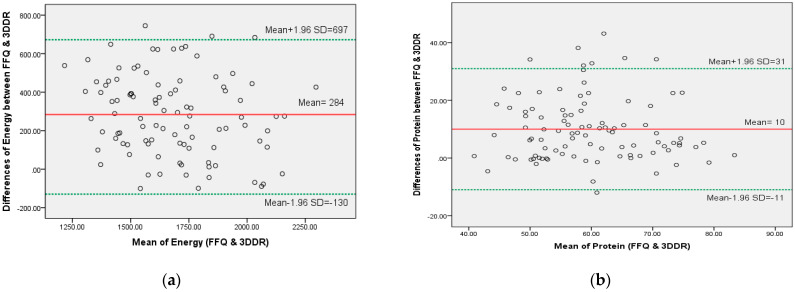
Bland-Altman plots visualizing the agreement between the BDHD-FFQ and 3DDRs. Bland-Altman plots picturing the agreement between the BDHD-FFQ and 3DDRs to evaluate the intake of (**a**) energy, (**b**) protein, (**c**) carbohydrate, (**d**) fat, (**e**) sodium, (**f**) calcium, (**g**) potassium, (**h**) iron and (**i**) phosphate. For all plots, the middle red line represents the mean differences between the BDHD-FFQ and 3DDRs, and limits of agreement (green line) show the 95% confidence interval (mean ± 1.96 SD).

**Table 1 nutrients-13-04521-t001:** Respective food group scores of specialists and amateurs.

Food Groups	Mean Score ^a^		*p*-Value ^b^
	Specialists (*n* = 12) ^†^	Amateurs (*n* = 12)	
A. Cereals and products	4.47 ± 0.27	4.33 ± 0.36	0.31
B. Rice (cooked)	4.43 ± 0.36	4.38 ± 0.42	0.76
C. Fish, Shellfish and their products	4.33 ± 0.37	4.50 ± 0.29	0.23
D. Meat, Poultry and their products	4.33 ± 0.38	4.42 ± 0.41	0.61
E. Vegetables	4.43 ± 0.37	4.55 ± 0.47	0.51
F. Pulses, legumes and their products	4.37 ± 0.42	4.55 ± 0.40	0.28
G. Milk and Dairy products	4.52 ± 0.45	4.43 ± 0.42	0.64
H. Bakery and Sweets	4.37 ± 0.44	4.50 ± 0.36	0.43
I. Fruit	4.38 ± 0.35	4.45 ± 0.49	0.70
J. Snacks and Finger Foods	4.3 5± 0.41	4.45 ± 0.42	0.56
K. Traditional Pitha	4.30 ± 0.32	4.38 ± 0.45	0.61
L. Fast Food Chain	4.40 ± 0.41	4.17 ± 0.49	0.22
M. Soup	4.32 ± 0.45	4.43 ± 0.43	0.53
N. Noodles and Pasta	4.32 ± 0.40	4.28 ± 0.51	0.86
O. Beverages	4.47 ± 0.42	4.35 ± 0.45	0.52
P. Miscellaneous	4.37 ± 0.42	4.35 ± 0.44	0.93
Overall	4.38 ± 0.38	4.41 ± 0.42	0.56

ᵃ Mean score presented as mean ± SD. ᵇ Independent Sample *t*-test between groups, *p*-value < 0.05; ^†^ Specialists consist of six nephrologists and six nutritionists.

**Table 2 nutrients-13-04521-t002:** Socio-demographic characteristics and clinical data of Phase III patients.

Characteristics (*n* = 97)		*n* (%)	Mean ± SD
Gender	Male	49 (50.5)	
	Female	48 (49.5)	
Marital status	Single	09 (9.3)	
	Married	76 (78.4)	
	Others	12 (12.4)	
Age (years)			49.8 ± 12.3
	20 to 30 years	06 (6.2)	
	31 to 40 years	21 (21.6)	
	41 to 50 years	23 (23.7)	
	51 to 60 years	31 (32.0)	
	>60 years	16 (16.5)	
BMI (kg/m^2^)			23.2 ± 4.9
Educational level	No formal education	08 (8.2)	
	Primary Education	23 (23.7)	
	Secondary Education	37 (38.1)	
	Higher Secondary Education	29 (29.9)	
Income level (Bangladeshi Taka)	Very Poor (<10,000)	17 (17.5)	
	Poor (10,000–20,000)	24 (24.7)	
	Moderate (20,000–30,000)	26 (26.8)	
	High (≥30,000)	30 (30.9)	
Working status	Yes	26 (26.8)	
	No	71 (73.2)	
Dialysis frequency (weekly)	3 times per Week	61 (62.9)	
	2 times per Week	36 (37.1)	
Duration of dialysis (hours)			3.8 ± 0.27
Dialysis vintage (month)			36 (18.5–68.5) ^a^
Blood Pressure	Systolic BP		147.5 ± 20.8
	Diastolic BP		85.9 ± 13.4
Hypertension	Yes	73 (75.3)	
	No	24 (24.7)	
Diabetes	Yes	45 (46.4)	
	No	52 (53.6)	
Pre-dialysis creatinine (μmol/L)			916.1 ± 312.5
Pre-dialysis urea (mg/dL)			72.2 ± 28.2
Serum potassium (mmol/L)			4.8 ± 1.1
Serum phosphate (mg/dL)			5.1 ± 2.0

Data were collected from 97 MHD patients from three district dialysis centers in Bangladesh. Values were mean ± standard deviation or frequency, *n* or percentage (%). ^a^ Median (interquartile range).

**Table 3 nutrients-13-04521-t003:** Correlations of BDHD-FFQ with 3DDR methods according to nutrients.

Nutrients (Unit)	Gross		Adjusted for Energy		Adjusted for Gender, Age and BMI	
	*r*	*p*-Value	*r*	*p*-Value	*r*	*p*-Value
Energy (Kcal)	0.66	0.001	-	-	0.18	0.001
Protein (g)	0.48	0.001	0.26	0.001	0.19	0.001
Carbohydrate (g)	0.58	0.001	0.38	0.001	0.20	0.001
Fat (g)	0.48	0.001	0.25	0.001	0.18	0.001
Calcium (mg)	0.41	0.001	0.28	0.001	0.11	0.001
Iron (mg)	0.30	0.002	0.23	0.001	0.10	0.003
Sodium (mg)	0.38	0.001	0.34	0.001	0.10	0.002
Potassium (mg)	0.39	0.001	0.49	0.001	0.10	0.002
Phosphate (mg)	0.43	0.001	0.53	0.001	0.12	0.001

Intraclass correlation coefficient *p*-value < 0.05. Interpretation criteria: good outcome (*r* ≥ 0.50), acceptable outcome (*r* = 0.20–0.49), and poor outcome (*r* < 0.20) [44]. Abbreviations: Kcal = kilocalorie; g = gram; mg = milligram.

**Table 4 nutrients-13-04521-t004:** Correlation coefficients between BDHD-FFQ and 3DDR with serum renal profiles.

Renal Profile																
Nutrients	Serum Phosphate				Serum Potassium				Serum Creatinine				Serum Urea			
	BDHD-FFQ		3DDR		BDHD-FFQ		3DDR		BDHD-FFQ		3DDR		BDHD-FFQ		3DDR	
	*r*	*p* ^a^	*r*	*p* ^a^	*r*	*p* ^a^	*r*	*p* ^a^	*r*	*p* ^a^	*r*	*p* ^a^	*r*	*p* ^a^	*r*	*p* ^a^
Energy	0.43	0.00	0.18	ns	0.47	0.00	0.19	ns	0.56	0.00	0.19	ns	0.55	0.00	0.16	ns
Protein	0.59	0.00	0.16	ns	0.41	0.00	0.24	0.02	0.27	0.00	0.10	ns	0.65	0.00	0.05	ns
Carbohydrate	0.28	0.00	0.17	ns	0.31	0.00	0.18	ns	0.55	0.00	0.18	ns	0.44	0.00	0.18	ns
Fat	0.23	0.02	0.12	ns	0.39	0.00	0.08	ns	0.21	0.04	0.13	ns	0.21	0.04	0.08	ns
Sodium	0.27	0.00	0.07	ns	0.32	0.00	0.11	ns	0.32	0.00	0.15	ns	0.16	ns	0.15	ns
Calcium	0.18	ns	0.12	ns	0.13	ns	0.14	ns	0.18	ns	0.21	0.04	0.03	ns	0.08	ns
Potassium	0.27	0.00	0.13	ns	0.27	0.00	0.11	ns	0.12	ns	0.03	ns	0.35	0.00	0.12	ns
Iron	0.32	0.00	0.24	0.02	0.00	ns	0.19	ns	0.20	0.05	0.22	0.03	0.16	ns	0.09	ns
Phosphate	0.52	0.00	0.18	ns	0.41	0.00	0.25	0.02	0.33	0.00	0.15	ns	0.48	0.00	0.06	ns

^a^ Pearson’s correlation coefficient *p*-value < 0.05. Interpretation criteria: good outcome (*r* ≥ 0.50), acceptable outcome (*r* = 0.20–0.49), and poor outcome (*r* < 0.20) [44]. Abbreviations: BDHD-FFQ, Bangladeshi Hemodialysis-Food Frequency Questionnaire; ns, not significant; 3DDR, 3-day diet recall.

**Table 5 nutrients-13-04521-t005:** Mean dietary nutrient consumption and mean difference between BDHD-FFQ and 3DDR.

Dietary Intake (Unit)	BDHD-FFQ	3DDR	% Mean Difference ^a^	*p*-Value ^b^
	Mean (SD)	Mean (SD)		
Energy (Kcal/day)	1823 (236)	1539 (275)	18.4	<0.01
Protein (g/day)	65 (11)	55 (11)	18.3	<0.01
Carbohydrate (g/day)	255 (44)	217 (44)	17.8	<0.01
Fat (g/day)	56 (11)	47 (11)	20.0	<0.01
Sodium (mg/day)	2469 (615)	2037 (577)	21.2	<0.01
Calcium (mg/day)	409 (134)	369 (123)	11.0	<0.01
Potassium (mg/day)	1819 (357)	1531 (309)	18.8	<0.01
Iron (mg/day)	16 (5)	12 (4)	33.9	<0.01
Phosphate (mg/day)	961 (155)	772 (198)	24.6	<0.01

^a^ Percentage mean difference were individually calculated by using the equation (BDHD-FFQ–3DDR)/3DDR × 100. Interpretation criteria: good 0–10.9%; acceptable 11.0–20.0%; poor >20.0% [51]. ᵇ Paired sample *t*-test of BDHD-FFQ with 3DDR, *p*-value < 0.05. Interpretation criteria: good: *p*-value > 0.05; poor: *p*-value ≤0.05 [52]. Abbreviations: BDHD-FFQ, Bangladeshi Hemodialysis-Food Frequency Questionnaire; 3DDR, 3-day diet recall; Kcal/day = kilocalorie per day; g/day = gram per day; mg/day = milligram per day.

**Table 6 nutrients-13-04521-t006:** Agreement of cross-quartile classification and weighted Kappa between BDHD-FFQ and 3DDR.

Nutrients (Unit)	Same Tertile (%)	Adjacent Tertile (%)	Opposite Tertile (%) ^a^	Weighted Kappa (95% CI) ^b^
Energy (Kcal/day)	42.27	44.33	2.06	0.43 (0.30;0.55)
Protein (g/day)	41.24	45.36	7.22	0.36 (0.23; 0.50)
Carbohydrate (g/day)	36.08	44.33	4.12	0.30 (0.16; 0.43)
Fat (g/day)	41.24	37.11	4.12	0.32 (0.17; 0.47)
Sodium (mg/day)	31.96	46.39	5.15	0.24 (0.10; 0.38)
Calcium (mg/day)	31.96	43.30	5.15	0.21 (0.08; 0.35)
Potassium (mg/day)	30.93	47.42	5.15	0.23 (0.09; 0.37)
Iron (mg/day)	29.90	38.14	7.22	0.12 (0.02; 0.27)
Phosphate (mg/day)	35.05	45.36	3.09	0.30 (0.16; 0.43)

^a^ Interpretation criteria (% in opposite tertile): good: ≤10%, poor: >10% [44]. ^b^ Interpretation criteria: good: ≥0.61; acceptable: 0.20–0.59; poor: <0.20 [44]. Abbreviations: 3DDR, 3-day diet recall; BDHD-FFQ, Bangladeshi Hemodialysis-Food Frequency Questionnaire; CI, confidence interval.

**Table 7 nutrients-13-04521-t007:** Limit of agreement (LoA) and correlation between mean and mean differences.

Nutrients (Unit)	Bland-Altman Index				Bland-Altman—Correlation Coefficient	
	Mean Difference	95% LoA Lower Limit; Upper Limit	Within 95% LoA	% within LoA	Pearson Correlation (*r*)	*p*-Value ^a^
Energy (Kcal)	283.79	−130.04; 697.62	94/97	96.9%	−0.19	>0.05
Protein (g)	10.01	−11.50; 31.50	89/97	91.8%	0.06	>0.05
Carbohydrate (g)	38.50	−40.17; 117.17	94/97	96.9%	−0.40	<0.05
Fat (g)	9.35	−12.72; 31.42	93/97	95.8%	0.01	>0.05
Sodium (mg)	432.74	−865.45; 1730.93	93/97	95.8%	0.07	>0.05
Calcium (mg)	40.63	−233.46; 314.72	91/97	93.8%	0.09	>0.05
Iron (mg)	287.39	−432.32; 917.10	93/97	95.8%	0.21	<0.05
Potassium (mg)	4.07	−5.92; 14.07	94/97	96.9%	0.16	>0.05
Phosphate (mg)	189.64	−181.97; 561.26	92/97	94.8%	−0.26	<0.05

^a^*p*-value for Pearson’s correlation coefficient. Interpretation criteria: good: *p*-value > 0.05; poor: *p*-value ≤ 0.05 [44].

## Data Availability

The datasets generated and analyzed for the current study are available from the corresponding author, Z.A.M.D., upon reasonable request.

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
