# Peer review of "A Food Frequency Questionnaire for Hemodialysis Patients in Bangladesh (BDHD-FFQ): Development and Validation"

_nutrients, 2021, doi:10.3390/nu13124521_

Round 1

Reviewer 1 Report

The manuscript entitled „ A Food Frequency Questionnaire for Hemodialysis Patients in Bangladesh (BDHD-FFQ): Development and Validation” presents interesting issue but some problems must be corrected.

Major:

  1. The applied method of validation and concluding is improper. For the validation of food frequency questionnaires, there are a specific recommendations (Cade et al, https://www.ncbi.nlm.nih.gov/pubmed/19079912), that should had been applied.
  2. According to the recommendations of Cade, the assessment of the new FFQ should include the assessment of reproducibility (that was not conducted in the presented study) and of validity. The assessment of reproducibility should be conducted and added to the manuscript.
  3. According to the recommendations of Cade, the specific methods should be applied in the validation studies. The analysis of correlation is not the recommended method (so Authors should not conclude on the basis of it). At the same time, the kappa statistic and Bland-Altman method are the recommended methods. Authors should present mainly the results of the recommended methods (they should be presented within the main body of the manuscript, instead of supplementary materials).
  4. While Authors used the Bland-Altman method, they applied it improperly. They should calculate the Bland-Altman index (in %) and conclude on the basis of the commonly indicated criteria (e.g. presented by Myles & Cui, https://www.ncbi.nlm.nih.gov/pubmed/17702826).
  5. Authors should use a proper scientific language, as they are preparing scientific manuscript, not a column for a newspaper, so some statements are inappropriate (e.g. “Chronic kidney disease (CKD) is one of the world’s leading silent killers”)

Abstract:

Authors should clearly present what was done and should focus on the most important results.

Introduction:

Authors should properly justify their study – present what is so far known based on the other studies for the construction of FFQs (what are possible methods of development, how are various FFQs constructed, etc.).

Materials and Methods:

Authors should clearly present what was done – what were the stages of their study, ho were they conducted with all necessary details.

Results:

Authors should focus on the most important results (including Bland-Altman method – as described above)

Authors should conclude about “acceptable validity” based on the Bland-Altman index rather than based on the other not recommended methods.

Discussion:

Authors should compare the data obtained for their FFQ with the results of validation obtained by other authors for other FFQs (various questionnaires for various populations) – in order to be able to conclude properly

Conclusions:

Authors should conclude based on the recommended methods of validation.

Minor:

 The manuscript should be prepared according to the instructions for authors (e.g. References)

Authors Contribution:

It seems that contribution of some Authors was only minor (HUR, TK) and they did not participate in preparing manuscript. There is a serious risk of a guest authorship procedure which is forbidden. In such case they should be rather presented in Acknowledgements Section and not be indicated as authors of the study.

Reviewer 2 Report

Figure 1 - Add the duration period of each phase.

Line 99 - what is the time interval of the 3 DDR? Was it 2 days of the week and one of the weekend?

Line 110 - what is implausible data to define in the methodology in addition to adding an organization chart for excluding participants and the reasons

Line 122 - that information can supplement line 99.

Two aspects to consider: reminders to generate the food list and reminders for the validation of the instrument. Do you both follow the same protocols?

- What is the duration of the study? It Mentioned February to April 2021 to recruit patients. Once recruited, the instrument validation process begins, then what would be the period that ended.

- Time interval between the application of recalls? Did you seek to contemplate the seasons of the year to assess variability of consumption?

- The FFQ was applied how long after the recalls?

 - Was the gross mean of the 3DDRs used or was the attenuation performed using the PC-side?

 - explain the energy adjustment process.

- did you do any reproducibility analysis of the FFQs?

- why not use intraclass correlation instead of pearson correlation?

Phase 1 - 102 patients with a total of 606 DDR, however, if there were 3 DDRs, wouldn't they be 306 DDR?

Table S5 does not measure relative validity and does make a plausibility assessment of consumption using the goldenberg technique. Correct the title on line 293.

Table 2 - does not have data on BMI, Hours of dialysis

Table 3 - good correlations are values ​​greater than 0.7. should in methodology add the classification of the Pearson correlation values. In this case, the results presented in adjusted data are considered with a low correlation.

Table 4 - 3DDRs always presented very low correlations compared to biomarkers.

The biochemical data were collected from medical records. What which time interval between the biochemical study and the date of obtaining the dietary data?

Table 6 - add the kappa statistic to evaluate concordances.

Figure S1 - The conclusion must be reviewed because the agreement is not acceptable, as some correlations are <0.3 considered a low correlation (must consider the adjusted values)

Round 2

Reviewer 1 Report

The manuscript entitled „A Food Frequency Questionnaire for Hemodialysis Patients in Bangladesh (BDHD-FFQ): Development and Validation” presents interesting issue but some problems must be corrected.

Major:

The applied method of validation and concluding is improper. For the validation of food frequency questionnaires, there are a specific recommendations (Cade et al, https://www.ncbi.nlm.nih.gov/pubmed/19079912), that should had been applied. If there are specific recommendations developed, Authors should not refer a single study (not being a recommendation/ guidelines/ statement) and base on it instead of using commonly applied recommendations.

According to the recommendations of Cade, the assessment of the new FFQ should include the assessment of reproducibility (that was not conducted in the presented study) and of validity. The assessment of reproducibility should be conducted and added to the manuscript. Authors should deeply discuss this problem and address it – without it stating that “A future study will address this aspect.” Indicates that Authors did not conduct their validation properly and they present partial results which do not justify conclusions.

While Authors used the Bland-Altman method, they applied it improperly. They should conclude based on the Bland-Altman index (in %) and commonly indicated criteria (e.g. presented by Myles & Cui, https://www.ncbi.nlm.nih.gov/pubmed/17702826). They should indicate the nutrients of result over 95% - energy, carbohydrate, Fe, K, as only for them the results indicated valid results.

Abstract:

Authors should focus on the most important results.

Introduction:

Authors should properly justify their study – present what is so far known based on the other studies for the construction of FFQs (what are possible methods of development, how are various FFQs constructed, etc.).

Results:

Authors should focus on the most important results (including Bland-Altman method – as described above)

Authors should conclude about “acceptable validity” based on the Bland-Altman index rather than based on the other not recommended methods.

Conclusions:

Authors should conclude based on the recommended methods of validation.

Reviewer 2 Report

Some minors to include in the methodology.
In phase 3, include how many recalls were evaluated at the end (figure and text)
Include the time interval for the application of r24h of phase 3  ( every 7 days?)

Author Response

The following are point-by-point response to the reviewer's comment: 

Q: In phase 3, include how many recalls were evaluated at the end (figure and text). 
A: We take note on this comments. We have added the following statement in the text (line 219-223) and they are highlighted in green. The number of recalls used in final analysis of Phase III has also been added in Figure 1.

“A  total of 348 24-hour recalls were collected from 116 patients recruited in the Phase III. However, only data from 97 patients (291 24-hour recalls) were used in the validation study. A total of 19 patients were excluded from the final analysis due to incomplete data (demographic/ biochemical/ BDHD-FFQ) (n=16) and dietary data under-reporting (n=3).”

Q: Include the time interval for the application of r24h of phase 3  (every 7 days?)

A: We take note on this comment. We have added the following statement in the text (Line 228-232)“ The 3DDR data comprise 24-hour dietary recalls on a weekend, and two weekdays (a dialysis day and a non-dialysis day). The time interval for the collection of 3DDR in the Phase III of the study is within a week (1-3 days between recalls depending on dialysis treatment schedule).